# Role of Non-Coding RNAs in Diagnosis, Prediction and Prognosis of Multiple Myeloma

**DOI:** 10.3390/cancers16051033

**Published:** 2024-03-02

**Authors:** Maciej Dubaj, Karol Bigosiński, Aleksandra Dembowska, Radosław Mlak, Aneta Szudy-Szczyrek, Teresa Małecka-Massalska, Iwona Homa-Mlak

**Affiliations:** 1Student Scientific Group, Department of Human Physiology, Medical University of Lublin, 20-080 Lublin, Poland; 56632@student.umlub.pl (K.B.); 55204@student.umlub.pl (A.D.); 2Department of Laboratory Diagnostics, Medical University of Lublin, Doktora Witolda Chodźki 1 Str., 20-093 Lublin, Poland; radoslawmlak@umlub.pl; 3Department of Hematooncology and Bone Marrow Transplantation, Medical University of Lublin, 20-081 Lublin, Poland; aneta.szudy-szczyrek@umlub.pl; 4Department of Human Physiology, Medical University of Lublin, 20-080 Lublin, Poland; teresa.malecka-massalska@umlub.pl (T.M.-M.); iwona.homa-mlak@umlub.pl (I.H.-M.)

**Keywords:** multiple myeloma, cancer, diagnosis, prognosis, miRNAs

## Abstract

**Simple Summary:**

The cornerstone of successful treatment of hematologic diseases, including multiple myeloma (MM), is early diagnosis and establishing a prognosis for the patient. Therefore, it is necessary to identify sensitive and specific markers for this disease. Non-coding RNAs including miRNAs are increasingly being recognized as potential diagnostic, predictive, and prognostic markers. These molecules are non-coding, single-stranded RNAs that regulate the expression of many target genes involved in key biological processes, such as cell proliferation, differentiation, and apoptosis. This paper aims to identify the role of non-coding RNAs including miRNAs as potential markers for diagnosis and prognosis in patients with MM.

**Abstract:**

Multiple myeloma (MM) is the second most common hematologic malignancy in the world and accounts for 15% of primary hemocytopathies, with an ever-increasing number of new cases. It is asymptomatic in 30% of instances; hence, the determination of highly sensitive and specific markers is necessary to make a proper diagnosis. In the last 20 years, miRNAs, involved in regulating the expression of genes responsible for cell proliferation and differentiation, including tumor cells, have been identified as potential diagnostic and prognostic markers. The main aim of the following review was to outline the role of miRNAs in the diagnosis and prognosis of MM, considering their role in the pathogenesis of the disease and identifying their target genes and pathways. For this purpose, publications dating from 2013–2023 have been reviewed. Based on the available data, it is concluded that non-coding RNAs including miRNAs could be potential markers in MM. Furthermore, they may serve as therapeutic targets for certain drugs.

## 1. Introduction

### 1.1. Multiple Myeloma

Multiple myeloma (MM), also known as plasma cell myeloma, is a cancer of the plasma cells of bone marrow [1]. MM accounts for 1% of all cancers and is the second most common hematological cancer (after lymphoma) [2]. Since 1990, the incidence of this disease has increased by as much as 120%, and approximately 200,000 cases are now reported annually around the world. MM is a disease mainly of older people (average age of diagnosis is 69 years); it occurs 1.5 times more often in men than in women and 2 times more often in African-Americans than in Caucasians [3]. The main risk factors for MM include: obesity, chronic inflammation (including in the course of cardiovascular diseases, diabetes and rheumatoid arthritis), exposure to pesticides, aromatic hydrocarbon solvents and ionizing radiation, as well as a positive family history [1,4]. The pathogenesis of MM is still not fully understood. Genetic abnormalities in plasma cells are found in almost all patients (90%) [1]. The vast majority of primary disorders (>90%) involve translocations between chromosome 14 (*IGH* gene, 14q32.33) and chromosomes 4 (*FGFR3*, *MMSET* genes, in 15% of MM cases), 11 (*CCND1*, in 15% MM), 16 (*MAF*, in 5% MM) and 20 (*MAFB*, in 1% MM). Moreover, 45% of patients have trisomies involving chromosomes 3, 5, 7, 9, 11, 15, 19 and/or 21. The following secondary genetic changes are observed: monosomy of chromosomes 13, 14, 17 (15–50% of patients with MM), translocations involving the *MYC* gene: t(8,14); t(8,11)—in 15% of patients and in 10–40% of them—deletions including the genes: *CDKN2C* and *FAM46C* (1p), *CD27* (12p), *RB1* and *DIS3* (13p), *TRAF3* (14p), *TP53* (17p). Moreover, in some patients (6–25%) mutations of the *BRAF*, *KRAS* and *NRAS* genes are observed, the products of which are involved in the MAPK signaling pathway, thus stimulating cell proliferation and survival [5]. One of the clinical conditions diagnosed before full-blown MM is monoclonal gammopathy of undetermined significance (MGUS). It is found when a monoclonal protein is present, without meeting the other diagnostic criteria for MM, including the percentage of plasmocytes in the bone marrow. The risk of MGUS progressing to MM is 1% per each year. MM could begin with an asymptomatic, pre-cancerous stage, so-called “smoldering” MM (sMM), which in 10% of patients progresses to full-blown MM within the first 5 years after diagnosis [6,7,8]. The main pathogenetic factors are three pathways: the replacement of normal bone marrow cells by cancer cells (resulting in a reduction in peripheral blood cell levels), the production of monoclonal protein (leading to kidney damage and thrombotic complications) and the secretion of inflammatory cytokines (causing, among others, the stimulation of osteoclasts and increased bone resorption) [1,9]. The above processes result in hypercalcemia, renal failure, anemia, bone pain and pathological fractures (so-called CRAB symptoms characteristic of MM). Additionally, recurrent infections and excessive bleeding occur in MM due to impaired immune system function and platelet dysfunction. Less specific symptoms include nausea, vomiting, general weakness, chronic fatigue and weight loss [9]. The diagnosis of MM is based mainly on the clinical picture, supported by the results of radiological and laboratory tests. Unlike other cancers, MM is characterized by osteolytic changes, causing the above-mentioned symptoms such as bone pain or osteoporosis. This is quite a characteristic phenomenon in this group of patients, occurring in 80% of cases [10]. The final diagnosis requires a bone marrow biopsy. The presence of damage to extramedullary organs (CRAB symptoms) should also be assessed, which is confirmed by, among others, high serum calcium and creatinine levels [1,11]. Treatment of MM originally involved the use of dexamethasone or prednisone with melphalan. Currently, drug regimens containing appropriate combinations of dexamethasone, cyclophosphamide and immunomodulatory drugs (thalidomide, lenalidomide, pomalidomide), proteasome inhibitors (bortezomib, carfilzomib, ixazomib), chimeric T-cell therapy (CAR-T), monoclonal antibodies (daratumumab, elotuzumab, isatuximab, belantamab mafodotin) and targeted therapies (selinexor, venetoclax, panobinostat) are used. Moreover, an important element of MM treatment is autologous stem cell transplantation (ASCT) [12,13]. The median survival after treatment varies depending on the therapy used and ranges from 3–4 to even 6 years, and after ASCT it is extended to 8 years [1,12].

### 1.2. Diagnostic and Prognostic Approaches for MM

The main approaches used in the diagnosis of MM include clinical imaging, laboratory and imaging examinations and bone marrow biopsy. Among the findings suggesting the diagnosis in blood tests are normocytic normochromic anemia (one of the first detections), sometimes thrombocytopenia and leukopenia. The erythrocyte sedimentation rate (ESR) often remains high, even above 100 mm/h. Also seen are hypercalcemia (10% of patients with an initial diagnosis), elevated levels of creatinine, uric acid, β2-microglobulin and increased lactate dehydrogenase (LDH) activity. In the case of reasonable suspicion of MM, electrophoresis of blood plasma and urine should be performed to demonstrate the presence of monoclonal light chains secreted by cancer cells. Other findings are: hyperproteinemia and decreased levels of normal immunoglobulins. Bone marrow aspiration or a trephane biopsy note an increased percentage of monoclonal plasmocytes. In addition, this test allows the performing of immunophenotyping and cytogenetics. The use of imaging methods makes it possible to detect lytic bone lesions, determine the size of the neoplastic infiltration or adequately perform differentiation and staging [14]. The following criteria must be met to make a final diagnosis of MM: the presence of at least 10% clonal plasma cells on bone marrow examination or biopsy-confirmed plasmocytoma and the presence of at least one myeloma-defining event (MDE). MDEs include the previously mentioned CRAB features, clonal bone marrow plasma cells ≥ 60%, serum involved/uninvolved (above/in reference range) free light chain (FLC) ratio ≥ 100 (provided FLC level is ≥100 mg/dL) and more than one focal lesion on MRI that is at least 5 mm in size [12].

Some factors are also known to predict a worse prognosis at the time of diagnosis. Cytogenetic factors can be detected by fluorescence in situ hybridization (FISH). These include t(4;14), t(14;16), t(14;20), del(1p), del(17p), 1q21+. The gene mutation associated with shorter overall survival (OS) is only *TP53*, found in 5% of patients. Other factors for an unfavorable prognosis include advanced age, onset of renal dysfunction, cachexia, central nervous system (CNS) involvement, International Staging System (ISS) stage 3, high LDH activity and developing plasma cell leukemia (PCL) [15].

### 1.3. miRNA

microRNA (miRNA) is a non-coding, single-stranded RNA, consisting of approximately 18–25 nucleotides, which is responsible for regulating the expression of many target genes in charge of such biological processes as cell proliferation, differentiation and apoptosis. miRNA constitutes approximately 1% of the human genome [16]. The miRNA coding sequences are transcribed by RNA polymerase II (Pol II), generating primary miRNAs characterized by a modified nucleotide at the 5′ end and a polyadenylated one at the 3′ end [17]. Studies have shown that miRNAs regulate the expression of genes that control cancer development (both oncogenes and tumor suppressor genes). The main mechanisms of action of these molecules include stimulation of cell proliferation and migration as well as inhibition of apoptosis and autophagy processes by inducing the synthesis of growth factors and influencing signaling pathways, both intracellular and between cells [1,17]. Such conclusions were drawn 20 years ago, where a relationship was indicated between the reduced expression of miR-15 and miR-16 in B cells and the increase in the level of Bcl-2 protein together with inhibition of apoptosis of cancer cells in chronic lymphocytic leukemia (CLL) [18]. Over the years, an important role of miRNA in the pathogenesis of other cancers has also been demonstrated. Menon et al. collected research results regarding the role of dysregulation of miRNA expression in the development of, among others, Wilms’ tumor (reduced expression of the let-7 family), colorectal cancer (reduced expression of miR-9-1, miR-129-2, miR-137, miR-124), gastric cancer (reduced expression of miR-9), renal cell carcinoma (overexpression of miR-17, miR-20a, decreased expression of miR-9), thyroid cancer (overexpression of miR-17), Merkel cell cancer (miR-1246, miR-375, miR-182, miR-183, miR-205, miR-15b, miR-96, miR-34a) and breast and ovarian cancer (overexpression of miR-21) [18,19,20]. Moreover, the role of impaired miRNA expression in the pathogenesis of MM and their function as potential diagnostic, predictive and prognostic markers in the course of MM have also been suggested [21,22]. What is also interesting, the use of nanoplexes containing miR-34a and the inhibition of miR-221 and miR-138 are potential therapeutic targets for the treatment of MM [23]. The main aim of this study is to describe the role of miRNA in the diagnosis and prognosis of MM.

## 2. The Role of miRNA Expression in Diagnosis and Prognosis of MM

### 2.1. The Role of miRNA as a Potential Diagnostic Factor for MM

Al Masri et al. in 2005 performed the first study describing the role of changes in miRNA expression in MM. They showed the occurrence of reduced expression of miR-1, miR-15, miR-16, miR-124a, miR-125b, miR-133a in plasma cells from cell lines and blood samples of MM patients [24]. The above discovery initiated an era of research that demonstrated the role of miRNA in the pathogenesis of MM [1,21,22]. Moreover, due to the detection of miRNAs in body fluids, many researchers propose their use as biomarkers suitable for diagnosis, prognosis and monitoring the effectiveness of MM treatment [1,25]. This is possible thanks to their stability and correlation of the expression level with the presence of pathological conditions [25,26]. 

#### 2.1.1. Studies Showing the Role of Increased miRNA Expression

Jiang et al. demonstrated overexpression of miR-125b-5p and miR-490-3p in the plasma of MM patients (*n* = 35) compared to the plasma of healthy subjects (*n* = 20). They can be effectively used as diagnostic markers in differentiating MM patients from representatives of the healthy population (respectively: AUC = 0.954, AUC = 0.866) [27]. Overexpression of miR-125b plays its role in the development of MM mainly by reducing the level of *p53*, and thus by inhibiting cell apoptosis [28]. Increased expression of miR-125 also induces overexpression of miR-34a [28]. In turn, overexpression of miR-490-3p, by stimulating *ERGIC3* and inhibiting *PCBP1*, promotes the proliferation and migration of cancer cells and disturbs the body’s natural immune response, taking part in the development of many cancers, including MM [29].

In a study of 90 MM patients, Zhao et al. observed that they had significantly increased levels of miR-1246 in serum compared to a healthy control group (*n* = 30). The researchers showed that overexpression of miR-1246 enabled the differentiation of MM patients from healthy subjects (AUC = 0.952). The role of this molecule in the pathogenesis of MM is probably based on promoting the migration and invasion of cancer cells through inhibition of *CXCR4* [30].

Chen et al., in their study conducted on sera from 90 patients with MM and 40 healthy subjects, showed that the patients had a significantly increased level of miR-448 compared to the control group and the group in remission. The marker could significantly be used in differentiating MM patients from healthy subjects (AUC = 0.888). The effect of miR-448 on the pathogenesis of MM is unknown. However, the role of dysregulation of miR-448 expression in the pathogenesis of pancreatic cancer (reduced expression of miR-448 inhibits apoptosis and stimulates the proliferation of cancer cells by stimulating Rab2B) and colon cancer (reduced expression of miR-448 increases the expression of IDO1 protein and thus stimulates apoptosis of CD8+ lymphocytes, intensifying cancer progression) or Hodgkin’s lymphoma (overexpression of miR-448 increases the level of DCLK1 protein and thus increases the proliferation and invasion of cancer cells) [31,32].

Shen et al., examining the sera of MM patients (*n* = 71) and healthy subjects (*n* = 46), noticed overexpression of miR-4449 in the patient population and its usefulness in differentiating both study groups (AUC = 0.885) [33]. 

Li et al. conducted a study in a group with MM (*n* = 23), MGUS (*n* = 16) and a healthy control group (*n* = 18). They showed increased expression of miR-134-5p, miR-107 and miR-15a-5p, both among patients with MM and MGUS, and their usefulness in differentiating MM patients from healthy ones (respectively: AUC = 0.812, AUC = 0.766, AUC = 0.719) and MGUS patients from healthy controls (respectively: AUC = 0.719, AUC = 740, AUC = 726). Additionally, a statistically significant correlation was demonstrated between the expression of miR107 and miR-15a-5p in MM, miR107 and miR15a-5p, as well as miR134-5p with miR-107 and miR15a-5p in MGUS (respectively: rho = 0.864, rho = 853, rho = 0.956, rho = 824). In order to differentiate patients with MM and MGUS, the best results are achieved by combining the measurements of miR-107, miR-15a-5p and hemoglobin (AUC = 0.954) [34]. Previous studies have proven the significant role of the above miRNA molecules in the pathogenesis of MM. The targets of miR-134-5p are the *ITGB1* and *PIK3R1* genes, which are stimulated by this miRNA and, through the IGF-1 signaling pathway, participate in the proliferation of MM cells. In turn, miR-15a-5p stimulates proliferation and growth and inhibits apoptosis of MM cells by inactivating the p53 protein and stimulating the PI3K-Akt, MAPK signaling pathways and increasing the levels of cytokines such as IL-6, IGF-1 and VEGF [32]. 

#### 2.1.2. Studies Showing the Role of Decreased miRNA Expression

Gupta et al. observed lower miR-203 expression in the serum of MM patients (*n* = 30) compared to healthy subjects (*n* = 30) and its potential in differentiating both groups (AUC = 0.930) [35]. The reduced expression of miR-203 in the pathogenesis of MM is mainly related to the stimulation of VCAN and, therefore, to the intensification of cell proliferation, adhesion and migration, the angiogenesis process and the weakening of the body’s immune response [35,36].

Li et al. showed that in the bone marrow of patients at the time of diagnosis (*n* = 90) the expression of miR-15a and miR-16-1 was significantly reduced compared to the marrow of healthy subjects (*n* = 19). The authors also noted that the levels of these two miRNAs were correlated (rho = 0.458). Reduced miR-15a and miR-16-1 expression can be used in differentiating patients with MM from healthy subjects (respectively: AUC = 0.664, AUC = 0.864) [37]. Reduced expression of the miR-15a/16-1 cluster has a proven role in the pathogenesis of MM. They regulate cell proliferation by inhibiting AKT3 kinase, MAP kinases, ribosomal protein S6 and the NF-kB activator MAP3KIP3 and influence angiogenesis and apoptosis by stimulating the expression of *VEGF*, *BCL2*, *CCND1*, *WNT3A* and *MCL1* genes [37].

In another study, Zhu et al. noticed that MM patients (*n* = 81) had significantly reduced miR-30d expression compared to healthy subjects (*n* = 78) and could be used to differentiate these groups (AUC = 0.800). Moreover, the authors indicated that reduced miR-30d expression causes activation of the PI3K/Akt pathway by stimulating the *MTDH* gene, inhibiting apoptosis and increasing the proliferation of cancer cells in MM [38].

Differences in miRNA levels in patients with sMM (*n* = 18) and MM (*n* = 76) in bone marrow aspirates were also described by Papanota et al. The authors showed that the levels of miR-16-5p and miR155-5p in CD38+ plasma cells were significantly reduced in the group of patients with full-blown MM compared to those in the sMM group [39].

Interestingly, not only serum or blood plasma can be a diagnostic material used to investigate differences in miRNA expression in MM patients. Li et al. indicated that it may also be urine. The authors observed that the levels of miR134-5p, miR6500-5p, miR-548q and miR-548y in urine were significantly lower in the group with newly diagnosed MM (*n* = 12) than in healthy subjects. A similar relationship was observed between the levels of the mentioned miRNAs in the group of MM patients in remission (*n* = 9) and healthy ones as well as in patients with relapse of MM after treatment (*n* = 6) and the control group. Moreover, they did not notice any statistically significant differences in the expression of the mentioned miRNAs between groups with newly diagnosed MM, in remission and relapse phases of the disease. Interestingly, they also noticed a negative correlation between the expression of the mentioned miRNAs and the total concentration of κ and λ light chains in urine, which are a recognized diagnostic marker of MM (respectively: rho = −0.427, rho = −0.461, rho = −0.469, rho = −0.493) [40]. Of the miRNAs described in the mentioned study, the role in the pathogenesis of MM is best known for miR-134-5p and the miR-548 family. The first of them stimulates the development of MM through the VEGF (increased angiogenesis) and JAK/STAT (disturbed apoptosis and cell cycle) signaling pathways. In turn, miRNAs from the miR-548 family stimulate 32 genes that regulate, among others: Ras, MAPK, PI3K/Akt and Hippo signaling pathways, taking part in the pathogenesis of MM and other cancers (renal clear cell carcinoma, breast cancer, nasopharyngeal cancer) [41]. 

#### 2.1.3. Studies Showing the Role of Both Increased and Decreased miRNA Expression

Kubiczkova et al., in a study in a group of 208 subjects (MM patients, n = 121, MGUS patients, n = 57, healthy subjects, n = 30), showed altered expression of five miRNAs in hematological patients compared to the healthy control group. The authors observed a statistically significant decrease in the expression of miR-744, miR-130a, let-7d and let-7e as well as an increase in the expression of miR-34a in the blood serum of patients with MM compared to the control group. Analogous changes were observed when comparing sera from patients with MGUS and the control group. Their analysis showed that miR-34a could be used in combination with let-7e to differentiate MM patients from healthy subjects (AUC = 0.898) and for MGUS patients (AUC = 0.976). Moreover, in blood serum collected from patients with relapsed MM (*n* = 18), a statistically significant increase in miR-34a expression and a decrease in let-7d expression was found compared to the results obtained at the time of MM diagnosis [26]. These results are in contradiction with the data presented in the review by Soliman et al., who indicated the role of reduced miR-34a expression in the pathogenesis of MM by inhibiting cell apoptosis and stimulating pro-survival signaling by affecting *CDK6*, *BCL2*, *TGIF2* and *NOTCH1* [1]. miR-130a molecules reduce C/EBP-ε expression during granulopoiesis, thereby increasing cell proliferation and inhibiting cell maturation. Moreover, it has been shown that miR-130a, through its inhibitory effect on *ATG2B* and *DICER1*, inhibits the survival of cancer cells in CLL. Therefore, its reduced expression may determine the progression of hematological diseases, including MM [42]. Reduced miR-744 expression, by stimulating the SOX2/Wnt/β-catenin signaling pathway, stimulates the proliferation, migration and invasion of MM cells, influencing the development and progression of cancer [43]. Overexpression of another described miRNA, let-7e, is associated with increased expression of the transcription factors PBX1 and CEBPA, which are involved in the pathogenesis of MM [44]. 

Hao et al., in a study in a group of 108 patients with MM and 40 healthy subjects, observed that the presence of cancer was associated with statistically significantly increased expression of miR-214 and miR-135b as well as decreased expression of miR-92a. Moreover, the authors proved that the overexpression of miR-214 and miR-135b was strongly positively correlated with the severity of osteolytic lesions in the course of MM (respectively: rho = 0.455, rho = 0.404). The aforementioned miRNAs can be used in the differentiation of patients with bone disease (respectively: AUC = 0.767, AUC = 0.907) [45]. The role of miR-135b in the pathogenesis of MM is to stimulate angiogenesis by inhibiting the production of the FIH1 factor [46]. In turn, overexpression of miR-214 is associated with a reduction in the level of PTEN protein and activation of the PI3K/Akt pathway, which leads to stimulation of uncontrolled cell proliferation. What is more, miR-214 stimulates osteoclastogenesis by stimulating the RANKL protein and stops the function and differentiation of osteoblasts by inhibiting the ATF4 transcription factor, which explains the association of this miRNA with bone changes in MM [45]. 

Zhang et al. compared the expression of miRNAs circulating in the serum of patients with clinically evident MM (*n* = 20), with sMM (*n* = 20) and in healthy subjects (*n* = 16). The authors found that subjects with MM had reduced let-7d-5p expression and miR-4741 overexpression compared to the healthy control group. In turn, in the sMM group, the level of miR103-3p was significantly lower and miR-4741 higher than in healthy subjects. Moreover, miR-103-3p, miR-140-3p and miR-425-5p were overexpressed, and miR-4505 was characterized by decreased expression in the MM group compared to the sMM group. In the pathogenesis of MM and the progression from sMM to MM, the let-7 family plays a special role, influencing, through the overexpression of the *MYC* gene, the intensification of cell proliferation and migration and disruption of their life cycle. In turn, miR-103 affects these cellular mechanisms by reducing the phosphorylation of the YAP protein and its involvement in the Hippo signaling pathway [47]. Detailed data on all mentioned studies are presented in Table 1. The involvement of the miRNAs through their targets in the pathogenesis of MM is shown in Figure 1. Figure 2 shows the changes in the expression levels of the various miRNAs in the diagnosis of MM depending on the material taken from the patient. A summary of AUC values of ROC curves in differentiating patients with active MM from healthy controls for individual miRNAs is presented in Figure 3. 

#### 2.1.4. Other Non-Coding RNA

What is also worth noting, other non-coding RNAs such as long non-coding RNA (lncRNA) and circular RNA (circRNA) are suggested as diagnostic markers in MM. Yang et al. compiled the results of studies conducted on serum of MM patients and healthy controls from 2017–2021, showing the statistically significant diagnostic value of overexpression of TUG1, PCAT1, H19, HOTAIR, LINC01606, PRINS, LBX2-AS1 and reduced expression of XLOC-013703 in differentiating these two groups [48]. Allegra et al. similarly collected data on circRNAs. They indicated that significant diagnostic potential in MM was found for decreased expression of AFF2, circMYBL2, circ_0000190, and increased expression of PTK2, CDYL, ATP10A, PVT1, circ_0000142, circ_0001821, circ_0069767, and circ_0007841 [49]. Zhu et al. additionally identified studies among MM patients in which significant diagnostic value was observed for overexpression of circ_101237, circ_001821, and circMYC, as well as decreased expression of circMYBL2 and circ_0087776 [50]. 

### 2.2. The Role of miRNAs as a Potential Predictive and Prognostic Factor for MM

It should be noted that to date there are no established molecular mechanisms nor predictive biomarkers for commonly used therapies in MM. However, it can be found in the literature that miRNAs, in addition to their previously described role as diagnostic markers, can also be used as predictive (allowing for the determination of the response to applied therapy and thus the selection of appropriate treatment options) and prognostic factors for MM.

#### 2.2.1. Studies Showing the Role of the Increased miRNA Expression

One of the prognostic factors described by Jiang et al. was miR-125b-5p overexpression. The median event-free survival (EFS) in patients with low expression of this miRNA was 13 months and in those with high expression 8 months [27].

Hao et al. observed that patients with higher miR-214 expression had shorter progression-free survival (PFS) (median, respectively: 8 vs. 22 months) and OS (median, respectively: 15 vs. 28 months). The unfavorable prognosis in this case is probably related to the activation of the PI3K/Akt pathway and the severity of uncontrolled proliferation of cancer cells [45].

Zhao et al. proved that miR-1246 overexpression was associated with an unfavorable prognosis, defined as a shortened PFS (median, respectively: 14 vs. 26.5 months) and OS (median, respectively: 20.5 vs. 55.5 months). This was probably related to the induction of extramedullary infiltration (through the inhibitory effect on *CXCR4*). Moreover, in pancreatic cancer and relapsed leukemia, high miR-1246 expression was shown to be associated with drug resistance to 5-fluorouracil and adriamycin by stimulating the miR-1246-AXIN2/GSK-3β-Wnt/β-catenin axis. Stimulation of this axis may also influence sensitivity to therapy in MM [30,51,52].

A similar relationship to Zhao et al. was observed by Ren et al. The authors observed that miR-1246 and miR-720 achieved higher expression in patients with MM (*n* = 60) compared to the healthy control group (*n* = 16). Moreover, the authors showed that their levels are correlated (rho = 0.621). What is more, they noticed that overexpression of both miRNAs in MM patients was associated with a significant shortening of PFS (median, respectively: 12.4 vs. 17.8 months) [53].

Szudy-Szczyrek et al., in a study in a group of 105 patients with MM, observed that high miR-8074 expression was associated with shorter PFS as well as OS (median, respectively: 17 vs. 39 months; 30 months vs. NR). The prognostic value of overexpression of this miRNA for shorter PFS and OS was also confirmed (respectively: HR = 2.28, 95%CI = 1.07–4.91; HR = 3.97, 95%CI = 1.32–11.90). Additionally, the authors noticed a positive correlation between miR-8074 and *MYC* expression (rho = 0.476), which may explain the role of this miRNA in the development of MM [54]. The *MYC* gene has a confirmed role in the pathogenesis of MM by intensifying cell proliferation and migration (Notch and Wnt signaling pathways) and stimulating angiogenesis (via VEGF), and its increased activity is associated with tumor progression and occurs in up to 60% of disease cases [55].

In a study on CD38+ plasma cells derived from bone marrow biopsies of MM patients (*n* = 86), Roseth Aass et al. proved that miR-105-5p overexpression was associated with shorter OS (1398 vs. 3313 days). This was confirmed in multivariate analysis (HR = 3.6, 95%CI = 1.56–8.5) [56]. The mechanism of miR-105-5p action in carcinogenesis was demonstrated in the case of gastric cancer, where the expression of this miRNA negatively correlated with the expression of PD-L1, stimulating the processes of apoptosis and tumor immunogenicity [57]. 

In another study, Papadimitriou et al. showed that increased miR-25-3p levels in the serum of MM patients (*n* = 69) may be a non-invasive marker of poor prognosis in this disease. It was associated with shorter OS (median, respectively: 25.8 vs. 30.7 months). This was confirmed by multivariate analysis (HR = 4.1, 95%CI = 0.79–21.32). A similar relationship was not observed for PFS [58]. The described miR-25-3p belongs to the miR-106b-25 cluster, which directly targets *TP53*, inhibiting its expression, and activates the PI3K/Akt signaling pathway, inducing *PTEN* expression, resulting in inhibition of apoptosis and cell aging [59]. Moreover, miR-25 overexpression influences MM resistance to dexamethasone by stimulating ULK1 and p27 (*CDKN1B*) [60]. 

#### 2.2.2. Studies Showing the Role of the Decreased miRNA Expression

In a previously described study, Kubiczkova et al. observed that reduced miR-744 and let-7e expression in MM patients was associated with a significantly lower one-year survival rate (respectively: 41.9% vs. 3.3% and 34.6% vs. 3.9%) and shorter time to progression (TTP) (median TTP respectively: 11.5 vs. 47.5 months and 11.5 vs. 47.5 months) [26]. The worse prognosis in the case of miR-744 dysregulation is related to the fact that it is located in the chromosomal region 17p12, and deletions from 17p13.1 (location of *TP53*) to 17p12 are known factors of worse prognosis in MM [42]. In turn, low levels of let-7 disrupt the cell cycle and stimulate MM cell proliferation by inhibiting genes such as *CCND1*, *MYC* and *RAS* [28].

Li et al. showed that patients with reduced miR-15a expression had statistically significantly shorter PFS (median, respectively: 14 vs. 29 months) and OS (median, respectively: 15 vs. 55 months) than patients with miR-15a overexpression [30]. The mechanisms of this phenomenon, apart from the above-described impact of the miR-15a/16-1 cluster on the cell cycle, also include the resistance it induces to bortezomib, one of the basic drugs used in MM therapy [37].

In their study, Manier et al. found that low levels of miR-18a and let-7b in the serum of MM patients (*n* = 156) strongly correlated with both shortened PFS (respectively: HR = 1.9, 95%CI = 1.22–2.94; HR = 2.76, 95%CI = 1.79–4.26) and OS (respectively: HR = 2.83, 95%CI = 1.07–7.5; HR = 4.52, 95%CI = 1.57–12.98), which makes them potential prognostic markers for MM. Low levels of let-7b, through the inhibition of *MYC*, *RAS* and *CCND1*, influence the induction of cell proliferation and growth, and thus the progression of cancer. In turn, miR-18a belongs to the miR-17-92 cluster, which naturally inhibits cancer development because it stimulates the expression of *IRF2* and thus regulates the immune response dependent on M1 macrophages and NK cells. Therefore, reducing the level of this miRNA is associated with a greater risk of rapid and more severe disease progression [61].

A similar relationship was reported for miR-19b and miR-331 by Navarro et al. In a study on sera from patients with MM (*n* = 33), MGUS (*n* = 8) and healthy subjects (*n* = 9), the authors indicated that reduced levels of these miRNAs are associated with a shorter PFS, and therefore a worse prognosis (median, respectively: 1.8 vs. 6 years; 2.9 vs. 8.6 years). Moreover, a combination of low expression of both miRNAs was identified as a prognostic factor associated with shorter PFS (HR = 5.3, 95%CI = 1.1–24.7) [62]. In turn, reduced expression of miR-331 stimulates *HDAC4*, increasing proliferation and disturbing the apoptosis of MM cells and resistance to bortezomib [63]. The significant role of reduced miR-331-3p expression on the stimulation of proliferation, migration and invasion of ovarian cancer cells by directly targeting and inhibiting *RCC2* was also confirmed [64]. Detailed data for all mentioned studies are presented in Table 2. The involvement of the miRNAs through their targets in the pathogenesis of poor survival in MM is shown in Figure 4. 

#### 2.2.3. Other Non-Coding RNA

Also interesting is the role of lncRNAs and circRNAs as potential markers of prediction and prognosis in MM. This includes increased expression of so-called “cancer-promoting lncRNAs” [48]. Yang et al. indicate that high expression of ANRIL, LOC606724, SNHG1, H19, NEAT1, HCP5, UCA1, PCAT1, PRAL, TCF7, BM742401, BDNF-AS, CRNDE, LINC01606, LUCAT1, LINC00461, OIP5-AS1, CCAT1, MIAT, RMRP is associated with shortened OS in patients. In addition, overexpression of PRAL and RMRP are associated with disease-free survival (DSF), while ANRIL, NEAT, HCP5, PCAT1, NR-046683, and ANGPTL1-3 are associated with PSF. Additionally, high expression of ANRIL, LOC606724, NEAT1, SNHG1, TCF7 and ANGPTL1-3 suggests a lower chance of achieving a clinical response (CR) [48]. Moreover, Carrasco-Leon et al. in a study of samples from 38 MM patients noted that overexpression of PDLIM1P4 and ENSG00000249988 is associated with shorter PFS and OS, while high expression of ENSG00000254343 and low expression of SMILO show an association with shorter PFS and OS, respectively [65]. According to a review by Allegra et al. as well as Zhu et al. higher levels of circ_0000190, circ_0069767 were associated with longer OS and PFS, while higher levels of circRNA_101237 were associated with shorter OS and PF and bortezomib resistance, similarly for circMYC. In contrast, overexpression of circ_0001821 was associated with lower OS in patients aged > 60 years [49,50].

Although miRNAs provide promising results, especially in the diagnosis and prognosis of MM, certain limitations in their use should be noted. First, there is no uniform protocol for miRNA collection, storage, and determination in MM, making direct comparison of results difficult. Moreover, many miRNAs show altered levels in non-cancer diseases, potentially leading to decreased specificity and fault-positive results. Moreover, the precise role of many miRNAs in the pathogenesis of MM is not fully understood. Therefore, the inclusion of miRNAs in MM diagnostic or therapeutic protocols requires a precise clarification of the relation between those molecules and their targets and standardization of methods of sample collection and analysis.

## 3. Conclusions

The studies cited in this review provide evidence that selected miRNAs might be useful diagnostic and prognostic biomarkers in MM. Moreover, selected miRNAs including miR-15a/16-1, miR-331, miR-125b-5p, and miR-214 may be useful in predicting resistance to commonly used drugs, including bortezomib or dexamethasone. Therefore, the determination of miRNA expression levels in MM may not only enable early diagnosis in a minimally invasive way but also improve clinical outcomes through personalized medicine based on biomarker-dependent qualification for treatment.

## Figures and Tables

**Figure 1 cancers-16-01033-f001:**
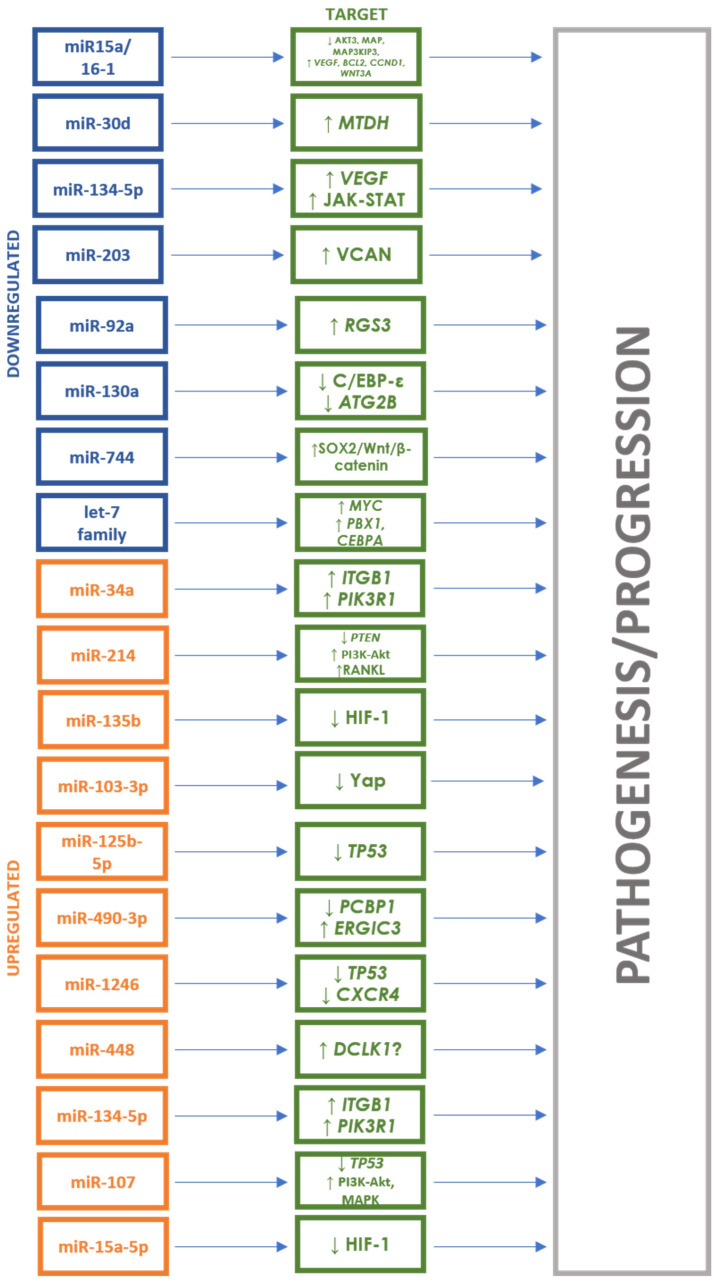
miRNAs and their targets in multiple myeloma diagnosis. ↑ = upregulation, ↓ = downregulation.

**Figure 2 cancers-16-01033-f002:**
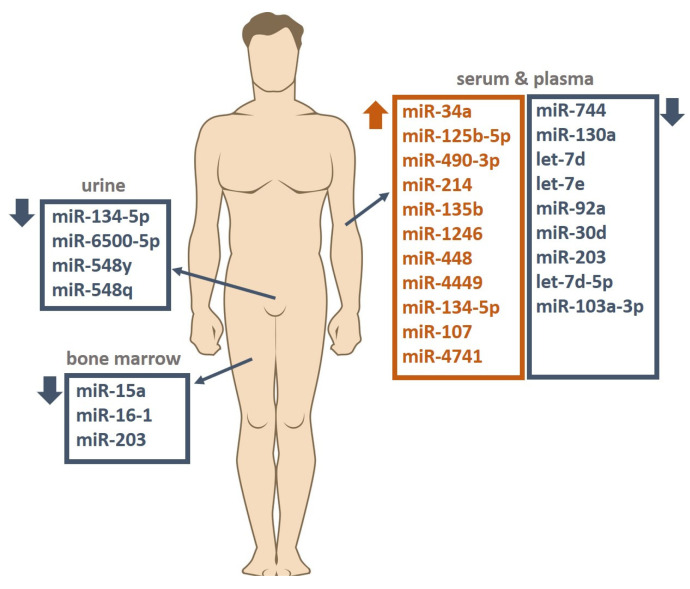
Changes in the expression levels of the various miRNAs in the diagnosis of multiple myeloma depending on the material taken from the patient. ↑ = upregulation, ↓ = downregulation.

**Figure 3 cancers-16-01033-f003:**
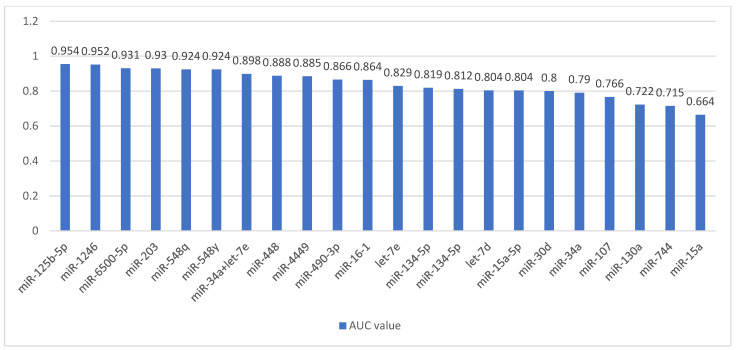
The bar chart showing area-under-the-curve values of ROC curves in differentiating patients with active MM from healthy controls for individual miRNAs in descending order.

**Figure 4 cancers-16-01033-f004:**
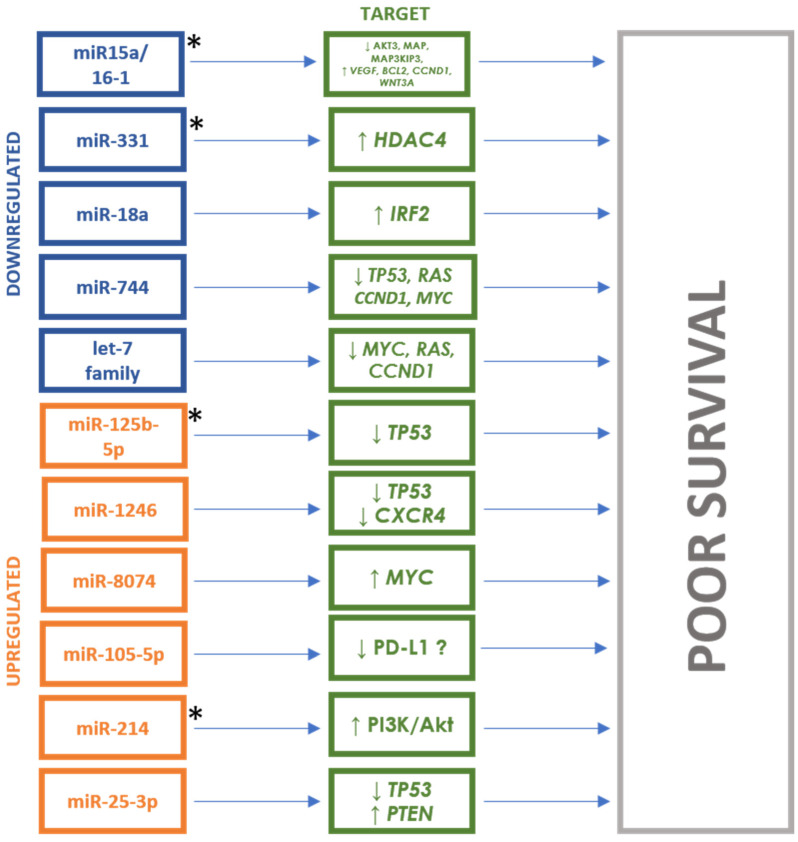
miRNAs and their targets in multiple myeloma prognosis. * = miRNAs potentially responsible for drug resistance. ↑ = upregulation, ↓ = downregulation.

**Table 1 cancers-16-01033-t001:** The role of miRNAs in diagnosis of multiple myeloma.

Authors	Study Group	Race/ Nationality	Study Material	Studied miRNA	miRNA Change	miRNA Targets
Kubiczkova et al. (2014) [26]	Newly diagnosed MM (*n* = 103), patients in relapse (*n* = 18), MGUS (*n* = 57), healthy donors (*n* = 30)	Czech	Serum	miR-744	Downregulated	*TUBB4*, *APC2*, *JUNB*
				miR-130a		*IGF1*, *CCND2*, *TGFβ*
				let-7d		*APC2*, *TGFβRI*, *CDC25A*, *TP53*
				let-7e		*MAPK6, IGF1*, *MYCN*, *CDK6*, *APC2*, *TP53*
				miR-34a	Upregulated	*MYCN*, *E2F3*, *BCL2*, *CDK6*
Jiang et al. (2018) [27]	MM (*n* = 35), healthy donors (*n* = 20)	China	Plasma	miR-125b-5p	Upregulated	*TP53*, *IRF4*
				miR-490-3p		*ERGIC3*, *PCBP1*
Zhao et al. (2022) [30]	MM (*n* = 90), healthy donors (*n* = 30)	China	Serum	miR-1246	Upregulated	*TP53*, *CXCR4*
Chen et al. (2022) [31]	Newly diagnosed MM (*n* = 30), patients in relapse (*n* = 30), patients in remission (*n* = 30), healthy donors (*n* = 40)	China	Serum	miR-448	Upregulated	-
Shen et al. (2017) [33]	Newly diagnosed MM (*n* = 71), healthy donors (*n* = 46)	China	Serum	miR-4449	Upregulated	-
Gupta et al. (2019) [35]	Newly diagnosed MM (*n* = 30), healthy donors (*n* = 30)	India	Bone marrow, serum	miR-203	Downregulated	*VCAN*
Li et al. (2020) [34]	MM (*n* = 23), MGUS (*n* = 16), healthy donors (*n* = 18)	China	Serum	miR-134-5p	Upregulated	*ITGB1, PIK3R1*
Li et al. (2015) [37]	Newly diagnosed MM (*n* = 90), patients in remission (*n* = 16), patients in relapse (*n* = 11), healthy donors (*n* = 19)	China	Bone marrow	miR-15a	Downregulated	*BCL2*, *MCL1, ETS1*, *JUN*, *TP53*
				miR-16-1		*BCL2*, *MCL1*, *CCND1*, *WNT3A*, *VEGF*
Zhu et al. (2018) [38]	Newly diagnosed MM (*n* = 81), healthy donors (*n* = 78)	China	Serum	miR-30d	Downregulated	*MTDH*
Li et al. (2020) [40]	Newly diagnosed MM (*n* = 12), patients in relapse (*n* = 6), patients in remission (*n* = 9), healthy donors (*n* = 12)	China	Serum, urine	miR-134-5p		*VEGF*, *JAK-STAT*, CML pathway
				miR-6500-5p	Downregulated	Lysosomal pathway
				miR-548q		*MAPK, RAS*, PI3K-Akt, Hippo
				miR-548y		*MAPK*, *RAS*, PI3K-Akt, Hippo
Hao et al. (2016) [45]	Newly diagnosed MM (*n* = 108), healthy donors (*n* = 44)	China	Serum	miR-214	Upregulated	*FBXW7*, *PTEN*, *AKT*, *GSK3*
				miR-135b		*GSK3*, FIH1
				miR-92a	Downregulated	*RGS3*
Zhang et al. (2019) [47]	MM (*n* = 20), sMM (*n* = 20), healthy donors (*n* = 16)	China	Serum	let-7d-5p	Downregulated	*MYC*
				miR-103a-3p		YAP, Hippo
				miR-4741	Upregulated	-

MM = multiple myeloma, MGUS = monoclonal gammopathy of undetermined significance.

**Table 2 cancers-16-01033-t002:** The role of miRNAs as predictive and prognostic factors in multiple myeloma.

Authors	Study Group	Race/ Nationality	Study Material	Assessed Study Endpoints (Associated Change)	Studied miRNA	miRNA Change	miRNA Targets
Kubiczkova et al. (2014) [26]	Newly diagnosed MM (*n* = 103), patients in relapse (*n* = 18), MGUS (*n* = 57), healthy donors (*n* = 30)	Czech	Serum	1-year survival rate (↓), TTP (↓)	miR-744	Downregulated	*TUBB4*, *APC2*, *JUNB*
					let-7e		*MAPK6*, *IGF1*, *MYCN*, *CDK6*, *APC2*, *TP53*
Jiang et al. (2018) [27]	MM (*n* = 35), healthy donors (*n* = 20)	China	Plasma	EFS (↓)	miR-125b-5p	Upregulated	*TP53*, *IRF4*
Zhao et al. (2022) [31]	MM (*n* = 90), healthy donors (*n* = 30)	China	Serum	PFS (↓), OS (↓)	miR-1246	Upregulated	*TP53*, *CXCR4*
Li et al. (2015) [37]	Newly diagnosed MM (*n* = 90), patients in remission (*n* = 16), patients in relapse (*n* = 11), healthy donors (*n* = 19)	China	Bone marrow	PFS (↓), OS (↓)	miR-15a	Downregulated	*BCL2*, *MCL1*, *ETS1*, *JUN*, *TP53*
					miR-16-1		*BCL2*, *MCL1*, *CCND1*, *WNT3A*, *VEGF*
Hao et al. (2016) [45]	Newly diagnosed MM (*n* = 108), healthy donors (*n* = 44)	China	Serum	PFS (↓), OS (↓)	miR-214	Upregulated	*FBXW7*, *PTEN*, *AKT*, *GSK3*
Ren et al. (2017) [53]	Newly diagnosed MM (*n* = 60), healthy donors (*n* = 16)	China	Serum	PFS (↓)	miR-720	Upregulated	*TP53*
					miR-1246		
Szudy-Szczyrek et al. (2022) [54]	MM (*n* = 105)	Poland	Serum	PFS (↓), OS (↓)	miR-8074	Upregulated	*TP53*, *MYC*, *MAPK1* *KIAA*
Roseth Aass et al. (2023) [56]	MM (*n* = 86)	Norway	Bone marrow	OS (↓)	miR-105-5p	Upregulated	PD-L1, F522
Papadimitriou et al. (2023) [58]	MM (*n* = 69)	Greece	Serum	OS (↓), PFS (ns)	miR-25-3p	Upregulated	*TP53*, *PTEN*, PI3K/Akt, *MYC*
Manier et al. (2017) [61]	Newly diagnosed MM (*n* = 156), healthy donors (*n* = 5)	France	Serum	PFS (↓), OS (↓)	miR-18a	Downregulated	*IRF2*
					let-7b		*MYC*, *RAS*, *CCND1*
Navarro et al. (2015) [62]	Newly diagnosed MM and after aHSCT (*n* = 33), MGUS (*n* = 8), healthy donors (*n* = 8)	Spain	Serum	PFS (↓)	miR-19b	Downregulated	*PTEN*, *IL6R*
					miR-331		*HDAC4*

MM = multiple myeloma, MGUS = monoclonal gammopathy of undetermined significance, TTP = time to progression, EFS = event-free survival, PFS = progression-free survival, OS = overall survival, ↓ = downregulation.

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
