# Peer review of "Role of Non-Coding RNAs in Diagnosis, Prediction and Prognosis of Multiple Myeloma"

_cancers, 2024, doi:10.3390/cancers16051033_

Round 1
Reviewer 1 Report
Comments and Suggestions for Authors
Below some comments and minor observations for improving this interesting review manuscript:
1. In this review manuscript, the authors comprehensively describe the clinical implication, mainly for diagnostic and prognostic purposes of miRNA dysregulations in multiple myeloma. The writing of this review manuscript should be improved. It is relatively well organized, and of adequate quality and has the merit to summarise a vast literature, helping readers to find all the relevant information in this topic. However, some improvement is necessary (please see below). The manuscript is in agreement with the scope of the Journal “Cancers MDPI”, but it requires some improvements to be evaluated for publication.
2. Before 1.2 section I suggests including a section describing the most important and clinically useful diagnostic and prognostic approaches for multiple myeloma which are currently employed in the clinical setting
3. Aditional non coding RNAs, such as non coding RNAs and ciruclr RNAs has been theorized as diagnostic and prognostic markers in multiple myeloma PMID: 37144499, PMID: 33597729 and PMID: 35406472 PMID: 36133810. This information and supporting references should be included.
4. The use of miRNAs as therapeutics in cancer has been explored, I suggest including a couple of words, please check this work for multiple myeloma https://www.mdpi.com/2075-4426/12/9/1428
5. I suggest replacing the word “study” with “review” (line 29)
6. miRNAs are also useful as diagnostic markers for the early detection and prognostications of non-melanoma skin tumors PMID: 36477874 and PMID: 37611429. This information and supporting references should be included (lines 110-120)
7. Please check the distance among lines as it varies in different review sections
8. I suggest using “subjects” instead of “people”, e.g., line 133
9. In order to improve the reading of the manuscript, the use of p values form other studies can be avoided. Numerous percentages (%) are reported too, their use can be mitigated. Alternatively, these details can be moved to tables
10. 2.1 and 202 sections are too long; I suggest splitting they up in various subsections. For instance, functional studies can be separated from observational studies
11. Please avoid s abbreviations in figure and table captions,
Comments on the Quality of English LanguageEnglish Language is good
Author Response
RESPONSE TO REVIEWER 1
Dear Reviewer,
Thank you very much for taking the time to review this manuscript. All your suggestions have helped to improve the substantive quality of our work and its readability. Please find the detailed responses below.
Comment 1. Before 1.2 section I suggests including a section describing the most important and clinically useful diagnostic and prognostic approaches for multiple myeloma which are currently employed in the clinical setting.
Response 1. Thank you for pointing this out. We have revised the MM diagnosis aspect of the introduction and added section 1.2. on useful diagnostic and prognostic approaches for this disease.
Comment 2. Aditional non coding RNAs, such as non coding RNAs and ciruclr RNAs has been theorized as diagnostic and prognostic markers in multiple myeloma PMID: 37144499, PMID: 33597729 and PMID: 35406472 PMID: 36133810. This information and supporting references should be included.
Response 2. Thank you for this valuable comment, it certainly adds significant value to the paper. We have added relevant information from your suggested publications at the end of sections 2.1 and 2.2 for diagnosis and prognosis in MM, respectively.
Comment 3. The use of miRNAs as therapeutics in cancer has been explored, I suggest including a couple of words, please check this work for multiple myeloma https://www.mdpi.com/2075-4426/12/9/1428.
Response 3. Agree, this is another important point to describe the use of miRNAs in MM. Accordingly, we have added information from the suggested publication in section 1.3 (lines 146-147).
Comment 4. I suggest replacing the word “study” with “review” (line 29).
Response 4. Thank you, we replaced the word "study" with "review".
Comment 5. miRNAs are also useful as diagnostic markers for the early detection and prognostications of non-melanoma skin tumors PMID: 36477874 and PMID: 37611429. This information and supporting references should be included (lines 110-120).
Response 5. Thank you for this suggestion, we did not reach such information during our review. We have added the relevant information in section 1.3 (lines 141-142).
Comment 6. Please check the distance among lines as it varies in different review sections.
Response 6. Thank you, we have changed the formatting to automatic, at this point all interlines throughout the work should be equal.
Comment 7. I suggest using “subjects” instead of “people”, e.g., line 133.
Respone 7. Agree, we have changed the words "people" in this usage to "subjects".
Comment 8. In order to improve the reading of the manuscript, the use of p values form other studies can be avoided. Numerous percentages (%) are reported too, their use can be mitigated. Alternatively, these details can be moved to tables.
Response 8. Thank you, we have removed all "p" values and % in parts 2.1 and 2.2 where possible.
Comment 9. 2.1 and 202 sections are too long; I suggest splitting they up in various subsections. For instance, functional studies can be separated from observational studies.
Response 9. Thank you for your comment. Agree, the form of the text actually may have made it difficult to read sections 2.1 and 2.2. We have introduced subsections, dividing the studies according to the levels of miRNA expression found in them. In addition, we have separated the individual studies by paragraphs, which now makes the paper easier to read.
Comment 10. Please avoid s abbreviations in figure and table captions.
Response 10. Thank you, we have removed the abbreviations in the captions of tables and figures.
Reviewer 2 Report
Comments and Suggestions for Authors
The authors conduct a review of the role of miRNAs in MM.
They focus on how miRNAs can be used for diagnosis, prognosis, as well prediction of therapy response.
Comments/suggestions to authors:
1-The manuscript needs re-writing to remove colloquial language.
2-The introduction of the disease would benefit from a MM MD expertise.
3-The structure of the manuscript is like a monolith of text. Hard to read through. A few diagrams summarizing the relationships between miRNAs and well-known MM drivers (e.g. mutations and cytogenetics) would be extremely helpful. Also, perhaps break the text by bullets or subtitles for each MM main mechanisms/drivers.
4-Some of the stats seem confusing or maybe have typos (see attached PDF for examples).
5-Conclusion is too light and generic.
Please see a few additional comments in the text (pdf).

Author Response
RESPONSE TO REVIEWER 2
Dear Reviewer,
Thank you very much for taking the time to read our article and evaluating it thoroughly. We have taken into consideration all comments and suggestions. Please read the responses below.
Comment 1. The manuscript needs re-writing to remove colloquial language.
Response 1. Thank you very much. We have re-written the article and corrected all the suggested colloquialisms.
Comment 2. The introduction of the disease would benefit from a MM MD expertise.
Response 2. The introduction has been reviewed by those working in the field of MM. Accordingly, we have improved the aspect of pathomechanisms of disease development and its diagnosis. In addition, we have clarified the difference between MGUS and sMM and refined the epidemiology of the risk of these forms progressing to full-blown MM.
Comment 3. The structure of the manuscript is like a monolith of text. Hard to read through. A few diagrams summarizing the relationships between miRNAs and well-known MM drivers (e.g. mutations and cytogenetics) would be extremely helpful. Also, perhaps break the text by bullets or subtitles for each MM main mechanisms/drivers.
Response 3. Thank you for your comment. Agree, the form of the text actually may have made it difficult to read sections 2.1 and 2.2. We have introduced subsections, dividing the studies according to the levels of miRNA expression found in them. In addition, we have separated the individual studies by paragraphs, which now makes the paper easier to read. We have also added the diagrams you suggested to sections 2.1. and 2.2.
Comment 4. Some of the stats seem confusing or maybe have typos (see attached PDF for examples).
Response 4. We checked all the statistics in the relevant references. Indeed, several of them were typos. However, one of them (line 404) is included this way in the cited paper, although it seems confusing.
Comment 5. Conclusion is too light and generic.
Response 5. Thank you for your comment. We have re-written the conclusion. We hope that now this section is relevant.
Comment 6. Please see a few additional comments in the text (pdf).
Response 6. Thank you for such precise suggestions and comments, all have been corrected.
Round 2
Reviewer 1 Report
Comments and Suggestions for Authors
The manuscript can be accepted in the present form
Author Response
Thank you for taking the time to evaluate the article, all the valuable suggestions that improved the quality of the paper and for the final approval.
Reviewer 2 Report
Comments and Suggestions for Authors
This revised version addressed most of the concerns directly pointed out by the reviewer, but I have the overall feeling that this is a rushed review and the manuscript would have benefited from an actual MM MD carefully reading through and editing the text. Since this is not my role, I will not list each sentence that needs re-writing, but provide my last pieces of advice, and suggest authors rework their manuscript.
A few important points of notice:
1-Figures 1 and 4 are a great summary of the review, but please clarify what each means: for example, do miRNAs of 1 promote progression, while the ones in Figure 4 promote multi-drug resistance?
2-Figure 3, please clarify these are AUCs of ROC curve, and what they are supposed to classify (e.g. healthy plasma cells vs. active MM?).
3-Section 1.1, between lines 62 and 73: The most natural order would be MGUS->SMOL->MM. Last sentence has ":" twice and confuses symptoms with pathways.
4-Section 1.2-Between lines 97 and 101: this uses a confusing term, money-string, which I could only find in a recent Chinese article. MM diagnosis is mainly based on M-proteins in blood/urine and bone marrow biopsy. Once again, need a MM MD to review these sections, as these do not appear to be proper guidelines.
5-Conclusion needs more work, as many sentences are generic and some are incorrect: "Moreover, some of them enable the prediction of resistance to drugs, including bortezomib or dexamethasone, at the time of diagnosis."). First, most samples are sensitive to these drugs at diagnosis, and second, there are no established molecular mechanisms nor predictive biomarkers for these therapies in MM.
Comments on the Quality of English LanguageComments on the writing of the text have been provided above.
